# BiGCN: A Bi-directional Low-Pass Filtering Graph Neural Network

## Abstract

Graph convolutional networks have achieved great success on graph-structured data. Many graph convolutional networks can be regarded as low-pass filters for graph signals. In this paper, we propose a new model, BiGCN, which represents a graph neural network as a bi-directional low-pass filter. Specifically, we not only consider the original graph structure information but also the latent correlation between features, thus BiGCN can filter the signals along with both the original graph and a latent feature-connection graph. Our model outperforms previous graph neural networks in the tasks of node classification and link prediction on most of the benchmark datasets, especially when we add noise to the node features.

## 1 Introduction

Graphs are important research objects in the field of machine learning as they are good carriers for structural data such as social networks and citation networks. Recently, graph neural networks (GNNs) received extensive attention due to their great performances in graph representation learning. A graph neural network takes node features and graph structure (e.g. adjacency matrix) as input, and embeds the graph into a lower-dimensional space. With the success of GNNs (Kipf & Welling, 2017; Veličković et al., 2017; Hamilton et al., 2017; Chen et al., 2018) in various domains, more and more efforts are focused on the reasons why GNNs are so powerful (Xu et al., 2019).

Li *et al* (Li et al., 2018) re-examined graph convolutional networks (GCNs) and connected it with Laplacian smoothing. NT and Maehara *et al* (NT & Maehara, 2019) revisited GCNs in terms of graph signal processing and explained that many graph convolutions can be considered as low-pass filters (e.g.(Kipf & Welling, 2017; Wu et al., 2019)) which can capture low-frequency components and remove some feature noise by making connective nodes more similar. In fact, these findings are not new. Since its first appearance in Bruna et al. (2014), spectral GCNs have been closely related to graph signal processing and denoising. The spectral graph convolutional operation is derived from Graph Fourier Transform, and the filter can be formulated as a function with respect to the graph Laplacian matrix, denoted as $g(L)$. In general spectral GCNs, the forward function is: $H^{(l+1)} = \sigma(g(L)H^{(l)})$.

Kipf and Welling (Kipf & Welling, 2017) approximated $g(L)$ using first-order Chebyshev polynomials, which can be simplified as multiplying the augmented normalized adjacency matrix to the feature matrix. Despite the efficiency, this first-order graph filter is found sensitive to changes in the graph signals and the underlying graph structure (Isufi et al., 2016; Bianchi et al., 2019). For instance, on isolated nodes or small single components of the graph, their denoising effect is quite limited due to the lack of reliable neighbors. The potential incorrect structure information will also constrain the power of GCNs and cause more negative impacts with deeper layers. As noisy/incorrect information is inevitable in real-world graph data, more powerful and robust GCNs are needed to solve this problem. In this work, we propose a new graph neural network with more powerful denoising effects from the perspective of graph signal processing and higher fault tolerance to the graph structure.

Different from image data, graph data usually has high dimensional features, and there may be some latent connection/correlation between each dimensions. Noting this, we take this connection information into account to offset the efforts of certain unreliable structure information, and remove extra noise by applying a smoothness assumption on such a "feature graph". Derived from the additional Laplacian smoothing regularization in this feature graph, we obtain a novel variant of

spectral GCNs, named BiGCN, which contains low-pass graph filters for both the original graph and a latent feature connection graph in each convolution layer. Our model can extract low-frequency components from both the graphs, so it is more expressive than the original spectral GCN; and it removes the noise from two directions, so it is also more robust.

We evaluate our model on two tasks: node classification and link prediction. In addition to the original graph data, in order to demonstrate the effectiveness of our model with respect to graph signal denoising and fault tolerance, we design three cases with noise/structure mistakes: randomly adding Gaussian noise with different variances to a certain percentage of nodes; adding different levels of Gaussian noise to the whole graph feature; and changing a certain percentage of connections. The remarkable performances of our model in these experiments verify our power and robustness on both clean data and noisy data.

The main contributions of this work are summarized below.

- We propose a new framework for the representation learning of graphs with node features. Instead of only considering the signals in the original graph, we take into account the feature correlations and make the model more robust.
- We formulate our graph neural network based on Laplacian smoothing and derive a bi-directional low-pass graph filter using the Alternating Direction Method of Multipliers (ADMM) algorithm.
- We set three cases to demonstrate the powerful denoising capacity and high fault tolerance of our model in tasks of node classification and link prediction.

## 2 RELATED WORK

We summarize the related work in the field of graph signal processing and denoising and recent work on spectral graph convolutional networks as follows.

### 2.1 GRAPH SIGNAL PROCESSING AND DENOISING

Graph-structured data is ubiquitous in the world. Graph signal processing (GSP) (Ortega et al., 2018) is intended for analyzing and processing the graph signals whose values are defined on the set of graph vertices. It can be seen as a bridge between classical signal processing and spectral graph theory. One line of the research in this area is the generalization of the Fourier transform to the graph domain and the development of powerful graph filters (Zhu & Rabbat, 2012; Isufi et al., 2016). It can be applied to various tasks, such as representation learning and denoising (Chen et al., 2014). More recently, the tools of GSP have been successfully used for the definition of spectral graph neural networks, making a strong connection between GSP and deep learning. In this work, we restart with the concepts from graph signal processing and define a new smoothing model for deep graph learning and graph denoising. It is worth mentioning that the concept of denoising/robustness in GSP is different from the defense/robustness against adversarial attacks (e.g. (Zügner & Günnemann, 2019)), so we do not make comparisons with those models.

### 2.2 SPECTRAL GRAPH CONVOLUTIONAL NETWORKS

Inspired by the success of convolutional neural networks in images and other Euclidean domains, the researcher also started to extend the power of deep learning to graphs. One of the earliest trends for defining the convolutional operation on graphs is the use of the Graph Fourier Transform and its definition in the spectral domain instead of the original spatial domain (Bruna et al., 2014). Defferrard *et al* (Defferrard et al., 2016) proposed ChebyNet which defines a filter as Chebyshev polynomials of the diagonal matrix of eigenvalues, which can be exactly localized in the k-hop neighborhood. Later on, Kipf and Welling (Kipf & Welling, 2017) simplified the Chebyshev filters using the first-order polynomial filter, which led to the well-known graph convolutional network. Recently, many new spectral graph filters have been developed. For example, the rational auto-regressive moving average graph filters (ARMA) (Isufi et al., 2016; Bianchi et al., 2019) are proposed to enhance the modeling capacity of GNNs. Compared to the polynomial ones, ARMA filters are more robust and provide a more flexible graph frequency response. Feedback-looped filters (Wijesinghe & Wang, 2019) further

improved localization and computational efficiency. There is also another type of graph convolutional networks that defines convolutional operations in the spatial domain by aggregating information from neighbors. The spatial types are not closely related to our work, so it is beyond the scope of our discussion. As we will discuss later, our model is closely related to spectral graph convolutional networks. We define our graph filter from the perspective of Laplacian smoothing, and then extend it not only to the original graph but also to a latent feature graph in order to improve the capacity and robustness of the model.

## 3 BACKGROUND: GRAPH SIGNAL PROCESSING

In this section, we will briefly introduce some concepts of graph signal processing (GSP), including graphs smoothness, graph Fourier Transform and graph filters, which will be used in later sections.

**Graph Laplacian and Smoothness.** A graph can be represented as $G = (V, E)$, which consists of a set of $n$ nodes $V = \{1, \ldots, n\}$ and a set of edges $E \subseteq V \times V$. In this paper, we only consider undirected attributed graphs. We denote the adjacency matrix of $G$ as $A = (a_{ij}) \in \mathbb{R}^{n \times n}$ and the degree matrix of $G$ as $D = diag(d(1), \ldots, d(n)) \in \mathbb{R}^{n \times n}$. In the degree matrix, $d(i)$ represents the degree of vertex $i \in V$. We consider that each vertex $i \in V$ associates a scalar $x(i) \in \mathbb{R}$ which is also called a graph signal. All graph signals can be represented by $x \in \mathbb{R}^n$. Some variants of graph Laplacian can be defined on graph $G$. We denote the graph Laplacian of $G$ as $L = D - A \in \mathbb{R}^{n \times n}$. It should be noted that the sum of rows of graph Laplacian $L$ is zero. The smoothness of a graph signal $x$ can be measure through the quadratic form of graph Laplacian: $\Delta(x) = x^T L x = \Sigma_{i,j} \frac{1}{2} a_{ij} (x(i) - x(j))^2$. Due to the fact that $x^T L x \geq 0$, $L$ is a semi-positive definite and symmetric matrix.

**Graph Fourier Transform and Graph Filters.** Decomposing the Laplacian matrix with $L = U \Lambda U^T$, we can get the orthogonal eigenvectors $U$ as Fourier basis and eigenvalues $\Lambda$ as graph frequencies. The Graph Fourier Transform $\mathcal{F} : \mathbb{R}^n \rightarrow \mathbb{R}^n$ is defined by $\mathcal{F}x = \hat{x} := U^T x$. The inverse Graph Fourier Transform is defined by $\mathcal{F}^{-1}\hat{x} = x := U\hat{x}$. It enables us to transfer the graph signal to the spectral domain, and then define a graph filter $g$ in the spectral domain for filtering the graph signal $x$:

$$g(L)x = Ug(\Lambda)U^T x = Ug(\Lambda)\mathcal{F}(x)$$

where $g(\Lambda) = diag(g(\lambda_1), \ldots g(\lambda_N))$ controls how the graph frequencies can be altered.

## 4 BIGCN

The Graph Fourier Transform has been successfully used to define various low-pass filters on graph signals (column vectors of feature matrix) and derive spectral graph convolutional networks (Defferrard et al., 2016; Bianchi et al., 2019; Wijesinghe & Wang, 2019). A spectral graph convolutional operation can be formulated as a function $g$ with respect to the Laplacian matrix $L$. Although it can smooth the graph and remove certain feature-wise noise by assimilating neighbor nodes, it is sensitive to node-wise noise and unreliable structure information. Notice that when the node features contain rich information, there may exist correlations between different dimensions of features which can be used to figure out the low-tolerance problem. Therefore, it is natural to define filters on "feature signals" (row vectors of graph feature matrix) based on the feature correlation. Inspired by this, we propose a bi-directional spectral GCN, named BiGCN, with column filters and row filters derived from the Laplacian smoothness assumption, as shown in Fig 1. In this way, we can enhance the denoising capacity and fault tolerance to graph structure of spectral graph convolutions. To explain it better, we start with the following simple case.

### 4.1 FROM LAPLACIAN SMOOTHING TO GRAPH CONVOLUTION

Assuming that $f = y_0 + \eta$ is an observation with noise $\eta$, to recover the true graph signal $y_0$, a natural optimization problem is given by:

$$\min_y \| y - f \|_2^2 + \lambda y^T L y,$$

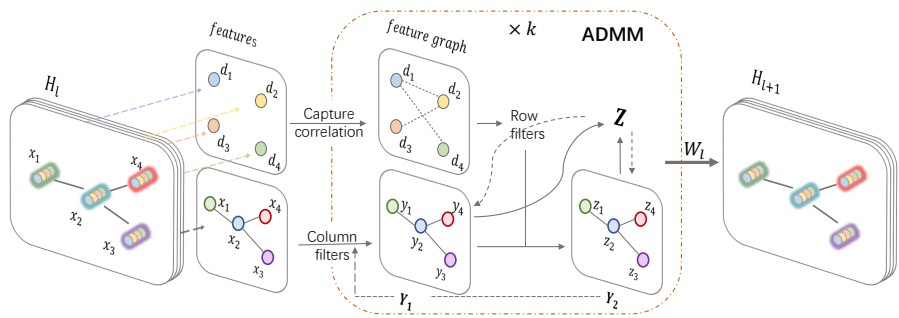

Figure 1: Illustration of one BiGCN layer. In the feature graph, $d_i$ indicates each dimension of features with a row vector of the input feature matrix as its "feature vector". We use a learnable matrix to capture feature correlations.

where $\lambda$ is a hyper-parameter, $L$ is the (normalized) Laplacian matrix. The optimal solution to this problem is the true graph signal given by

$$y = (I + \lambda L)^{-1} f. \tag{1}$$

If we generalize the noisy graph signal $f$ to a noisy feature matrix $F = Y_0 + N$, then the true graph feature matrix $Y_0$ can be estimated as follows:

$$Y_0 = \arg \min_{Y} \| Y - F \|_F^2 + \lambda trace(Y^T L Y) = (I + \lambda L)^{-1} F. \tag{2}$$

$Y^T L Y$, the Laplacian regularization, achieves a smoothness assumption on the feature matrix. $(I + \lambda L)^{-1}$ is equivalent to a low-pass filters in graph spectral domain which can remove feature-wise/column-wise noise and can be used to defined a new graph convolutional operation. Specifically, by multiplying a learnable matrix $W$ (i.e. adding a linear layer for node feature transformation beforehand, which is similar to (Wu et al., 2019; NT & Maehara, 2019)), we obtain a new graph convolutional layer as follows:

$$H^{(l+1)} = \sigma((I + \lambda L)^{-1} H^{(l)} W^{(l)}). \tag{3}$$

In order to reduce the computational complexity, we can simplify the propagation formulation by approximating $(I + \lambda L)^{-1}$ with its first-order Taylor expansion $I - \lambda L$.

## 4.2 BI-DIRECTIONAL SMOOTHING AND FILTERING

Considering the latent correlation between different dimensions of features, similar to the graph adjacency matrix, we can define a "feature adjacency matrix" $A'$ to indicate such feature connections. For instance, if $i - th, j - th, k - th$ dimension feature refer to "height", "weight" and "age" respectively, then "weight" may have very strong correlation with "height" but weak correlation with "age", so it is reasonable to assign $A'_{ji} = 1$ while $A'_{jk} = 0$ (if we assume $A'$ is a $0 - 1$ matrix). With a given "feature adjacency matrix", we can construct a corresponding "feature graph" in which nodes indicate each dimension of features and edges indicate the correlation relationship. In addition, if $Y_{n \times d}$ is the feature matrix of graph $G$, then $Y^T_{d \times n}$ would be the "feature matrix of the feature graph". That is, the column vectors of $Y_{n \times d}$ are the feature vectors of those original nodes while the row vectors are exactly the feature vectors of "feature nodes". Analogously, we can derive the Laplacian matrix $L'$ of this feature graph.

When noise is not only feature-wise but also node-wise, or when graph structure information is not completely reliable, it is beneficial to consider feature correlation information in order to recover the clean feature matrix better. Thus we add a Laplacian smoothness regularization on feature graph to the optimization problem indicated above:

$$\mathcal{L} = \min_{Y} \| Y - F \|_F^2 + \lambda_1 trace(Y^T L_1 Y) + \lambda_2 trace(Y L_2 Y^T). \tag{4}$$

Here $L_1$ and $L_2$ are the normalized Laplacian matrix of the original graph and feature graph, $\lambda_1$ and $\lambda_2$ are hyper-parameters of the two Laplacian regularization. $Y L' Y^T$ is the Laplacian regularization

on feature graph or row vectors of the original feature matrix. The solution of this optimization problem is equal to the solution of differential equation:

$$\frac{\partial \mathcal{L}}{\partial Y} = 2Y - 2F + 2\lambda_1 L_1 Y + 2\lambda_2 Y L_2 = 0. \tag{5}$$

This equation, equivalent to $\lambda_1 L_1 Y + \lambda_2 Y L_2 = F - Y$, is a Sylvester equation. The numerical solution of Sylvester equations can be calculated using some classical algorithm such as Bartels–Stewart algorithm (Bartels, 1972), Hessenberg-Schur method (Golub et al., 1979) and LAPACK algorithm (Anderson et al., 1999). However, all of them require Schur decomposition which including Householder transforms and QR iteration with $\mathcal{O}(n^3)$ computational cost. Consequently, we transform the original problem to a bi-criteria optimization problem with equality constraint instead of solving the Sylvester equation directly:

$$\mathcal{L} = \min_{Y_1} f(Y_1) + \min_{Y_2} g(Y_2) \quad s.t \ \ Y_2 - Y_1 = 0,$$

$$f(Y_1) = \frac{1}{2} \parallel Y_1 - F \parallel_F^2 + \lambda_1 trace(Y_1^T L_1 Y_1),$$

$$g(Y_2) = \frac{1}{2} \parallel Y_2 - F \parallel_F^2 + \lambda_2 trace(Y_2 L_2 Y_2^T). \tag{6}$$

We adopt the ADMM algorithm (Boyd et al., 2011) to solve this constrain convex optimization problem. The augmented Lagrangian function of $\mathcal{L}$ is:

$$\mathcal{L}_p(Y_1, Y_2, Z) = f(Y_1) + g(Y_2) + trace(Z^T(Y_2 - Y_1)) + \frac{p}{2} \parallel Y_2 - Y_1 \parallel_F^2 . \tag{7}$$

The update iteration form of ADMM algorithm is:

$$Y_1^{(k+1)} := arg \min_{Y_1} \mathcal{L}_p(Y_1, Y_2^{(k)}, Z^{(k)})$$

$$= arg \min_{Y_1} \frac{1}{2} \parallel Y_1 - F \parallel_F^2 + \lambda_1 trace(Y_1^T L_1 Y_1) + trace(Z^{(k)^T}(Y_2^{(k)} - Y_1)) + \frac{p}{2} \parallel Y_2^{(k)} - Y_1 \parallel_F^2,$$

$$Y_2^{(k+1)} := arg \min_{Y_2} \mathcal{L}_p(Y_1^{(k+1)}, Y_2, Z^{(k)})$$

$$= arg \min_{Y_2} \frac{1}{2} \parallel Y_2 - F \parallel_F^2 + \lambda_2 trace(Y_2 L_2 Y_2^T) + trace(Z^{(k)^T}(Y_2 - Y_1^{(k+1)}))$$

$$+ \frac{p}{2} \parallel Y_2 - Y_1^{(k+1)} \parallel_F^2,$$

$$Z^{(k+1)} = Z^{(k)} + p(Y_2^{(k+1)} - Y_1^{(k+1)}). \tag{8}$$

We obtain $Y_1$ and $Y_2$ iteration formulation by computing the stationary points of $\mathcal{L}_p(Y_1, Y_2^{(k)}, Z^{(k)})$ and $\mathcal{L}_p(Y_1^{(k+1)}, Y_2, Z^{(k)})$:

$$Y_1^{(k+1)} = \frac{1}{1+p}(I + \frac{2\lambda_1}{1+p} L_1)^{-1}(F + pY_2^{(k)} + Z^{(k)}),$$

$$Y_2^{(k+1)} = \frac{1}{1+p}(F + pY_1^{(k+1)} - Z^{(k)})(I + \frac{2\lambda_2}{1+p} L_2)^{-1}. \tag{9}$$

To decrease the complexity of computation, we can use first-order Taylor approximation to simplify the iteration formulations by choosing appropriate hyper-parameters $p$ and $\lambda_1, \lambda_2$ such that the eigenvalues of $\frac{2\lambda_1}{1+p} L_1$ and $\frac{2\lambda_2}{1+p} L_2$ all fall into $[-1, 1]$:

$$Y_1^{(k+1)} = \frac{1}{1+p}(I - \frac{2\lambda_1}{1+p} L_1)(F + pY_2^{(k)} + Z^{(k)}),$$

$$Y_2^{(k+1)} = \frac{1}{1+p}(F + pY_1^{(k+1)} - Z^{(k)})(I - \frac{2\lambda_2}{1+p} L_2),$$

$$Z^{(k+1)} = Z^{(k)} + p(Y_2^{(k+1)} - Y_1^{(k+1)}). \tag{10}$$

In each iteration, as shown in Fig 1, we update $Y_1$ by appling the column low-pass filter $I - \frac{2\lambda_1}{1+p}L_1$ to the previous $Y_2$, then update $Y_2$ by appling the row low-pass filter $I - \frac{2\lambda_2}{1+p}L_2$ to the new $Y_1$. To some extent, the new $Y_1$ is the low-frequency column components of the original $Y_2$ and the new $Y_2$ is the low-frequency row components of the new $Y_1$. After $k$ iteration (in our experiments, $k = 2$), we take the mean of $Y_1^{(k)}$ and $Y_2^{(k)}$ as the approximate solution $Y$, denote it as $Y = ADMM(F, L_1, L_2)$. In this way, the output of ADMM contains two kinds of low-frequency components. Moreover, we can generalize $L_2$ to a learnable symmetric matrix based on the original feature matrix $F$ (or some prior knowledge), since it is hard to give a quantitative description on feature correlations.

In $(l+1)^{th}$ propagation layer, $F = H^{(l)}$ is the output of $l^{th}$ layer, $L_2$ is a learnable symmetric matrix depending on $H^{(l)}$, for this we denote $L_2$ as $L_2^{(l)}$. The entire formulation is:

$$H^{(l+1)} = \sigma(ADMM(H^{(l)}, L_1, L_2^{(l)})W^{(l)}). \tag{11}$$

**Discussion about over-smoothing**    Since our algorithm is derived from a bidirectional smoothing, some may worry about the over-smoothing problem. The over-smoothing issue of GCN is explored in (Li et al., 2018; Oono & Suzuki, 2020), where the main claim is that when the GCN model goes very deep, it will encounter over-smoothing problem and lose its expressive power. From this perspective, our model will also be faced with the same problem when we stack many layers. However, a single BiGCN layer is just a more expressive and robust filter than a normal GCN layer. Actually, compared with the single-direction low-pass filtering GCN with a general forward function: $H^{(l+1)} = \sigma(g(L_1)H^{(l)}W^{(l)})$, $ADMM(H^{(l)}, L_1, L_2^{(l)})$, combining low-frequency components of both column and row vectors of $H^{(l)}$, is more informative than $g(L_1)H^{(l)}$ since the latter can be regarded as one part of the former to some extent. It also explains that BiGCN is more expressive that single-direction low-pass filtering GCNs. Furthermore, when we take $L_2$ as an identity matrix (in equation 5), BiGCN degenerates to a single-directional GCN with low-pass filter: $((1+\lambda_2)I + \lambda_1 L_1)^{-1}$. It also illustrates that BiGCN has more general model capacity. More technical details are added in Appendix.

In practice, we can also mix the BiGCN layer with original GCN layers or use jumping knowledge (Xu et al., 2018) to alleviate the over-smoothing problem: for example, we can use BiGCN at the bottom and then stack other GCN layers above. As we will show in experiments, the adding smoothing term in the BiGCN layers does not lead to over-smoothing; instead, it improves the performance on various datasets.

## 5 EXPERIMENT

We test BiGCN on two graph-based tasks: semi-supervised node classification and link prediction on several benchmarks. As these datasets are usually observed and carefully collected through a rigid screening, noise can be negligible. However, in many real-world data, noise is everywhere and cannot be ignored. To highlight the denoising capacity of the bi-directional filters, we design three cases and conduct extensive experiments on artificial noisy data. In noise level case, we add different levels of noise to the whole graph. In noise rate case, we randomly add noise to a part of nodes. Considering the potential unreliable connection on the graph, to fully verify the fault tolerance to structure information, we set structure mistakes case in which we will change graph structure. We compare our performance with several baselines including original GCN (Kipf & Welling, 2017), GraphSAGE (Hamilton et al., 2017), GAT (Veličković et al., 2017), GIN (Xu et al., 2019), and GDC (Klicpera et al., 2019).

### 5.1 BENCHMARK DATASETS

We conduct link prediction experiments on Citation networks and node classification experiments both on Citation networks and Co-purchase networks.

**Citation.** A citation network dataset consists of documents as nodes and citation links as directed edges. We use three undirected citation graph datasets: **Cora** (Sen et al., 2008), **CiteSeer** (Rossi & Ahmed, 2015) , and **PubMed** (Namata et al., 2012) for both node classification and link prediction tasks as they are common in all baseline approaches. In addition, we add another citation network **DBLP** (Pang et al., 2015) to link prediction tasks.

**Co-purchase.** We also use two Co-purchase networks **Amazon Computers** (McAuley et al., 2015) and **Amazon Photos** (Shchur et al., 2018), which take goods as nodes, to predict the respective product category of goods. The features are bag-of-words node features and the edges represent that two goods are frequently bought together.

## 5.2 EXPERIMENTAL SETUP

We train a two-layer BiGCN as the same as other baselines. Details of the hyperparameters setting and noise cases setting are contained in the appendix.

**Learnable $L_2$.** We introduce a completely learnable L2 in our experiments. In detail, we define $L_2 = I - D_2^{-1/2} A_2 D_2^{-1/2}$, $A_2 = W_2 + W_2^T$ where $W_2 = \text{sigmoid}(W)$ and $W$ is an uppertriangle matrix parameter to be optimized. To make it sparse, we also add L1 regularization to $L_2$. For each layer, $L_2$ is defined differently. Note that our framework is general and in practice there may be other reasonable choices for $L_2$ (e.g. as we discussed in Appendix).

## 5.3 BASELINE MODELS

We compare our BiGCN with several state-of-the-art GNN models: **GCN** (Kipf & Welling, 2017), **GraphSAGE** (Hamilton et al., 2017), **GAT** (Veličković et al., 2017), **GIN** (Xu et al., 2019): Graph Isomorphism Network, **GDC** (Klicpera et al., 2019): Graph diffusion convolution based on generalized graph diffusion. We compare one of the variants of GDC which leverages personalized PageRank graph diffusion to improve the original GCN and adapt GCN into link prediction tasks is consistent with the implementation in P-GNN.

## 5.4 RESULTS

We set three types of noise cases in terms of noise level, noise rate and structure mistake to evaluate each model on node classification and link prediction tasks (excluding structure mistakes). "Noise level" and "noise rate" add different types of noise to node features; "structure mistake" indicates we randomly remove or add edges in the original graph. For noise on node features, we expect our BiGCN show its ability as graph filters. For structural errors, we expect the latent feature graph can help with the correction of structural errors in original graphs. The detailed settings of these cases as well as some additional experimental results can be found in the Appendix.

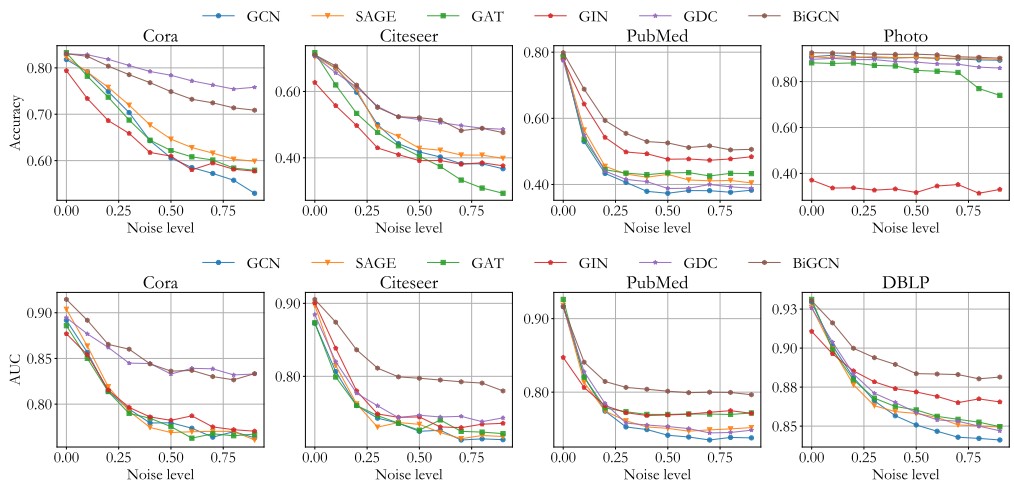

Figure 2: Node classification (Top) and link prediction (Bottom) results of models in noise level case.

**Noise level case.** In this case, we add Gaussian noise with a fixed variance (from 0.1 to 0.9, called the noise level) to the feature matrix. As Fig 2 shows, BiGCN outperforms other baselines and

shows flatter declines with increasing noise levels, demonstrating better robustness in both node classification and link prediction tasks.

**Noise rate case.** Here, we randomly choose a part of nodes at a fixed percentage (from 0.1 to 0.9, called the noise rate) to add different Gaussian noise. From Fig 3 we can see that, on the two tasks, BiGCN performs much better than baselines on all benchmarks apart from Cora. Especially on the PubMed dataset, BiGCN improves node classification accuracy by more than 10%.

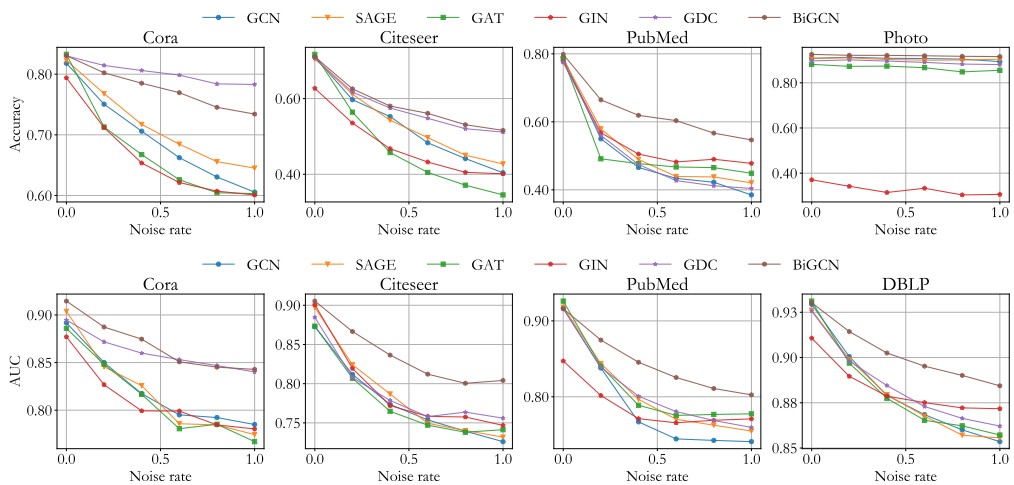

Figure 3: Node classification (Top) and link prediction (Bottom) results of models in noise rate case.

**Structure mistakes case.** Structure mistakes refer to the incorrect interaction relationship among nodes. In this setting, we artificially remove or add a certain percentage of edges of graphs at random and conduct experiments on node classification. Fig 4 illustrates the outstanding robustness of BiGCN that is superior to all baselines, demonstrating that our bi-directional filters can effectively utilize information from the latent feature graph and drastically reduce the negative impact of the incorrect structural information. At last, we would like to mention that our model also outperform other models in most cases on clean data without noise. This can attribute to BiGCN's ability to efficiently extract graph features through its bidirectional filters. The detailed values in the figures are listed in Appendix.

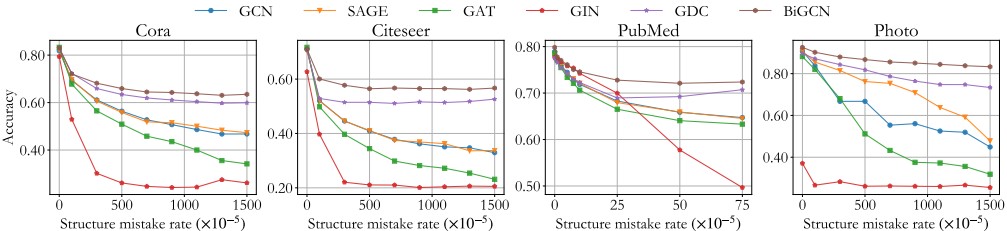

Figure 4: Node classification accuracy of models in structure mistakes case.

## 6    CONCLUSION

We proposed bidirectional low-pass filtering GCN, a more powerful and robust network than general spectral GCNs. The bidirectional filter of BiGCN can capture more informative graph signal components than the single-directional one. With the help of latent feature correlation, BiGCN also enhances the network's tolerance to noisy graph signals and unreliable edge connections. Extensive experiments show that our model achieves remarkable performance improvement on noisy graphs.

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

## A    MODEL EXPRESSIVENESS

In this section, we add more details about the our discussion of over-smoothing in Section 4.

As a bi-directional low-pass filter, our model can extract more informative features from the spectral domain. To simplify the analysis, let us take just one step of ADMM (k=1). Since $Z^0 = 0, Y_1^0 = Y_2^0 = F$, we have the final solution from Equation (10) as follows

$$Y_1 = (I - \frac{2\lambda_1}{1 + p}L_1)F,$$

$$Y_2 = (I - \frac{2p\lambda_1}{(1 + p)^2}L_1)F(I - \frac{2\lambda_2}{1 + p}L_2) = \left((I - \frac{2\lambda_2}{1 + p}L_2)F^T(I - \frac{2p\lambda_1}{(1 + p)^2}L_1)\right)^T.$$

From this solution, we can see that $Y_1$ is a low-pass filter which extracts low-frequency features from the original graph via $L_1$; $Y_2$ is a low-pass filter which extracts low-frequency features from the feature graph via $L_2$ and then do some transformation. Since we take the average of $Y_1$ and $Y_2$ as the output of $ADMM(H, L_1, L_2)$, the BiGCN layer will extract low-frequency features from both the graphs. That means, our model adds new information from the latent feature graph while not losing any features in the original graph. Compared to the original single-directional GCN, our model has more informative features and is more powerful in representation.

When we take more than one step of ADMM, from Equation (10) we know that the additive component $(I - \frac{2\lambda_1}{1 + p}L_1)F$ is always in $Y_1$ (with a scaling coefficient), and the component $F(I - \frac{2\lambda_2}{1 + p}L_2)$ is always in $Y_2$. So, the output of the BiGCN layer will always contain the low-frequency features from the original graph and the feature graph with some additional features with transformation, which can give us the same conclusion as the one step case.

## B    SENSITIVITY ANALYSIS

To demonstrate how hyper-parameters (iterations of ADMM, $\lambda_2$, $p$ and $\lambda$) influence BiGCN, we take Cora as an example and present the results on node classification under certain settings of artificial noise.

First, we investigate the influence of iteration and $\lambda_2$ on clean data and three noise cases with 0.2 noise rate, 0.2 noise level and 0.1% structure mistakes respectively. Fig 5(a) shows that ADMM with 2 iterations is good enough and the choice of $\lambda_2$ has very little impact on results since it can be absorbed into the learnable $L_2$. Then we take a particular case in which noise rate equals to 0.2 as an example to illustrate how much the performance of BiGCN depends on $p$ and $\lambda$. Fig 5(b) shows that $p$ guarantees relatively stable performance over a wide range values and only $\lambda$ has comparable larger impact.

## C    FLEXIBLE SELECTION OF $L_2$

In our paper, we assume the latent feature graph $L_2$ as a learnable matrix and automatically optimize it. However, in practice it can also be defined as other fixed forms. For example, a common way to deal with the latent correlation is to use a correlation graph Li et al. (2017). Another special case is if we define $L_2$ as an identity matrix, our model will degenerate to a normal (single-directional) low-pass filtering GCN. When we take $L_2 = I$ in Equation (5), the solution becomes

$$Y = ((1 + \lambda_2)I + \lambda_1 L_1)^{-1}F$$

which is similar to the single-directional low pass filter (Equation (2)). Then the BiGCN layer will degenerate to the GCN layer as follows:

$$H^{(l+1)} = \sigma(((1 + \lambda_2)I + \lambda_1 L_1)^{-1}H^{(l)}W^{(l)}).$$

To show the difference between different definitions of $L_2$, we design a simple approach using a thresholded correlation matrix for $L_2$ to compare with the method used in our main paper. In particular, we define an edge weight $A_{ij}$ as follows.

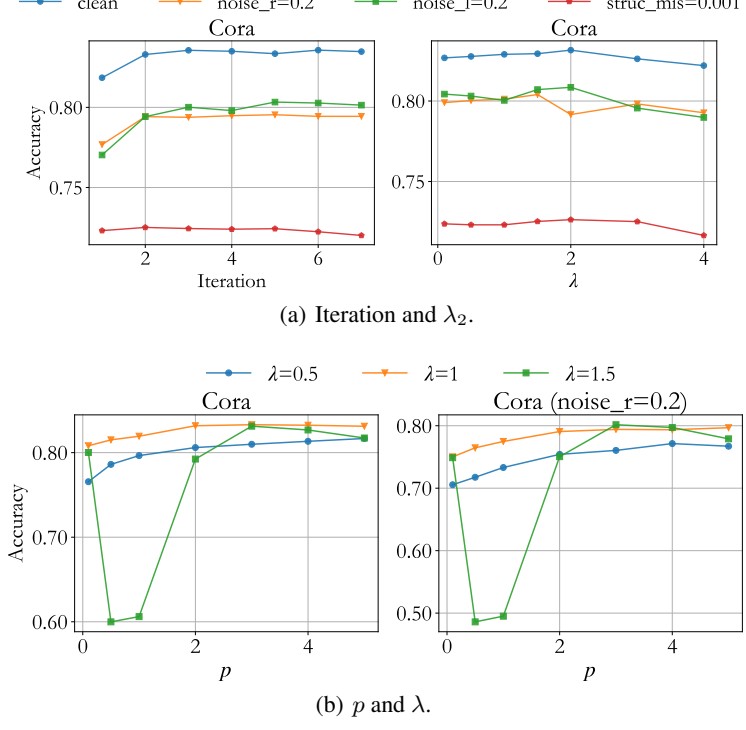

(a) Iteration and $\lambda_2$.

(b) $p$ and $\lambda$.

Figure 5: Sensitivity analysis of iteration, $\lambda_2$, $\lambda$ and $p$ on node classification. For iteration and $\lambda_2$, we conduct experiments on clean data and three noise cases with 0.2 noise rate, 0.2 noise level and 0.1% structure mistakes respectively. For $p$ and $\lambda$, we provide the performance of BiGCN on Cora with 0.2 noise rate.

$$(P_{ij})_{j\in\mathcal{N}(i)\cup i} = softmax([\frac{x_i^T x_j}{\parallel x_i \parallel \parallel x_j \parallel}]_{j\in\mathcal{N}(i)\cup i}),$$

$$A_{ij} = \begin{cases} 0, & P_{ij} \le mean(P) \\ 1, & P_{ij} > mean(P) \end{cases}.$$

Then we compute $L_2$ as the normalized Laplacian obtained from $A$, i.e. $L_2 = \tilde{D}^{-\frac{1}{2}}\tilde{A}\tilde{D}^{-\frac{1}{2}}$. For a simple demonstration, we only compare the two models on Cora with node feature noises. From Table 1 and Table 2, we can see that our learnable $L_2$ is overall better. However, a fixed $L_2$ can still give us decent results. When the node feature dimension is large, fixing $L_2$ may be more efficient.

Table 1: Node classification accuracy in noise rate case on Cora dataset of two types of L2.

| Noise_rate | 0.200 | 0.400 | 0.600 | 0.800 | 1.000 |
|---|---|---|---|---|---|
| Fixed_$L_2$ | 0.807 | 0.774 | 0.756 | 0.733 | 0.726 |
| Learnable_$L_2$ | 0.802 | 0.785 | 0.770 | 0.745 | 0.734 |

Table 2: Node classification accuracy in noise level case on Cora dataset of two types of L2.

| Noise level | 0.100 | 0.200 | 0.300 | 0.400 | 0.500 | 0.600 | 0.700 | 0.800 | 0.900 |
|---|---|---|---|---|---|---|---|---|---|
| Fixed_$L_2$ | 0.823 | 0.804 | 0.777 | 0.753 | 0.725 | 0.713 | 0.702 | 0.696 | 0.691 |
| Learnable_$L_2$ | 0.825 | 0.804 | 0.785 | 0.768 | 0.749 | 0.732 | 0.725 | 0.714 | 0.709 |

## D EXPERIMENTAL DETAILS

We train a two-layer BiGCN as the same as other baselines using Adam as the optimization method with 0.01 learning rate, $5 \times 10^{-4}$ weight decay, and 0.5 dropout rate for all benchmarks and baselines. In the node classification task, we use early stopping with patience 100 to early stop the model training process and select the best performing models based on validation set accuracy. In the link prediction task, we use the maximum 100 epochs to train each classifier and report the test ROCAUC selected based on the best validation set ROCAUC every 10 epochs. In addition, we follow the experimental setting from P-GNN (position-aware GNN) and the approach that we adapt GCN into link prediction tasks is consistent with the implementation in P-GNN. We set the random seed for each run and we take mean test results for 10 runs to report the performances.

All the experimental datasets are taken from PyTorch Geometric and we test BiGCN and other baselines on the whole graph while in GDC, only the largest connected component of the graph is selected. Thus, the experimental results we reported of GDC maybe not completely consistent with that reported by GDC. We found that the Citation datasets in PyTorch Geometric are a little different from those used in GCN, GraphSAGE, and GAT. It may be the reason why their accuracy results on Citeseer and Pubmed in node classification tasks are a little lower than the original papers reported.

To highlight the denoising capacity of the bi-directional filters, we design the following three cases and conduct extensive experiments on artificial noisy data. The noise level case and noise rate cases are adding noise on node features and the structure mistake case adds noise to graph structures.

**Noise level case.** In this case, we add different Gaussian noise with zero mean to all the node features in the graph, i.e. to the feature matrix and use the variance of Gaussian (from 0.1 to 0.9) as the quantitative indexes of noise level.

**Noise rate case.** In this case, we add Gaussian noise with the same distribution to different proportions of nodes, i.e. some rows of the feature matrix, at a random and quantitatively study how the percentage (from 10% to 100%) of nodes with noisy features impacts the model performances.

**Structure mistakes case.** In practice, it is common and inevitable to observe wrong or interference link information in real-world data, especially in a large-scale network, such as a social network. Therefore, we artificially make random changes in the graph structure, such as removing edges or adding false edges by directly reversing the value of the original adjacency matrix (from 0 to 1 or from 1 to 0) symmetrically to obtain an error adjacency matrix. We choose different scales of errors to decide how many values would be reversed randomly. For example, assigning a 0.01% error rate to a graph consisting of 300 vertices means that $0.01 \times 10^{-2} \times 300^2 = 9$ values symmetrically distributed in the adjacency matrix will be changed.

We conduct all of the above cases on five benchmarks in node classification tasks and the two previous cases on four benchmarks in link prediction tasks.

For more experimental details please refer to our codes: `https://anonymous.4open.science/r/4fefefed-4d59-4214-a324-832ac0ef1e96/`.

### D.1 DATASETS

We use three Citation networks (Cora, Citeseer, and Pubmed) and two Co-purchase networks for node classification tasks and all the Citation datasets for link prediction.

Table 3: Bechmark Dataset.

| Dataset | Type | Nodes | Edges | Features | Classes | Label Rate |
|---|---|---|---|---|---|---|
| Cora | Citation | 2,708 | 5,278 | 1,433 | 7 | 0.052 |
| Citeseer | Citation | 3,327 | 4,552 | 3,703 | 6 | 0.036 |
| Pubmed | Citation | 19,717 | 44,324 | 500 | 3 | 0.003 |
| DBLP | Citation | 17716 | 105734 | 1639 | 4 | / |
| AMZ Comp | Co-purchase | 13,752 | 245,861 | 767 | 10 | 0.015 |
| AMZ Photos | Co-purchase | 7,650 | 119,081 | 745 | 8 | 0.021 |

## D.2 EXPERIMENTAL RESULTS ON CLEAN DATA

The performances of models on clean benchmarks in node classification and link prediction are shown in Table 4 and 5 respectively. These results correspond to the values with noise level 0 in the figures of Section 5.

Table 4: BiGCN compared to GNNs on node classification tasks, measured in accuracy (%). Standard deviation errors are given.

|       | Cora | Citeseer | PubMed | Comp | Photo |
|-------|------|----------|--------|------|-------|
| GCN   | $81.8 \pm 0.6$ | $71.0 \pm 0.6$ | $78.9 \pm 0.6$ | $82.7 \pm 4.6$ | $90.8 \pm 1.3$ |
| SAGE  | $82.3 \pm 0.5$ | $70.5 \pm 0.7$ | $78.5 \pm 0.5$ | $83.1 \pm 4.2$ | $90.8 \pm 1.1$ |
| GAT   | $\mathbf{83.1} \pm 0.5$ | $\mathbf{71.7} \pm 0.5$ | $78.5 \pm 0.5$ | $76.3 \pm 3.5$ | $88.2 \pm 1.3$ |
| GIN   | $79.4 \pm 0.8$ | $62.7 \pm 1.2$ | $77.7 \pm 0.7$ | $41.4 \pm 3.6$ | $37.1 \pm 12.0$ |
| GDC   | $83.0 \pm 0.6$ | $70.7 \pm 0.7$ | $77.5 \pm 0.6$ | $84.5 \pm 0.8$ | $89.7 \pm 0.4$ |
| BiGCN | $\mathbf{83.1} \pm 0.7$ | $71.0 \pm 0.6$ | $\mathbf{80.0} \pm 0.3$ | $\mathbf{87.0} \pm 0.6$ | $\mathbf{92.6} \pm 0.3$ |

Table 5: BiGCN compared to GNNs on link prediction tasks, measured in ROC AUC (%). Standard deviation errors are given.

|       | Cora | Citeseer | PubMed | DBLP |
|-------|------|----------|--------|------|
| GCN   | $89.2 \pm 0.8$ | $87.3 \pm 1.7$ | $91.7 \pm 0.8$ | $92.9 \pm 0.4$ |
| SAGE  | $90.4 \pm 0.7$ | $89.7 \pm 0.7$ | $91.8 \pm 0.3$ | $92.6 \pm 0.2$ |
| GAT   | $88.6 \pm 0.8$ | $87.3 \pm 1.1$ | $\mathbf{92.6} \pm 0.4$ | $\mathbf{93.1} \pm 0.3$ |
| GIN   | $87.7 \pm 0.7$ | $90.1 \pm 1.3$ | $84.7 \pm 0.6$ | $91.1 \pm 0.4$ |
| GDC   | $89.5 \pm 0.4$ | $88.5 \pm 1.1$ | $91.6 \pm 0.7$ | $92.6 \pm 0.4$ |
| BiGCN | $\mathbf{91.5} \pm 0.5$ | $\mathbf{90.5} \pm 0.7$ | $91.6 \pm 0.3$ | $\mathbf{93.1} \pm 0.3$ |

## D.3 EXPERIMENTAL RESULTS ON AMZ COMP

The node classification performances of models on AMZ Comp dataset are shown in Fig 6.

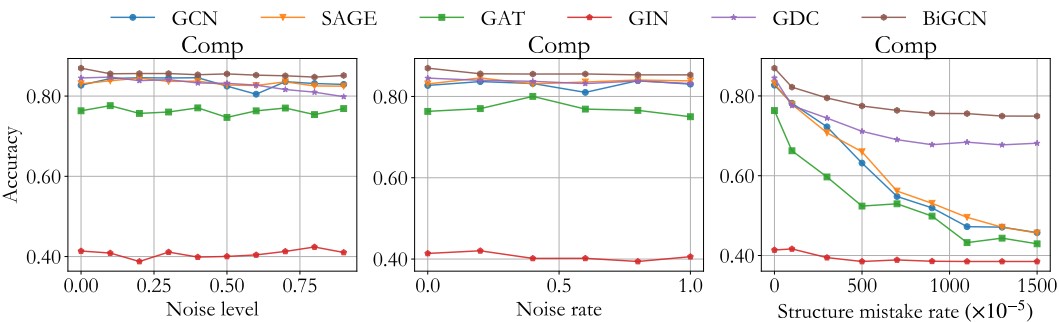

Figure 6: Node classification accuracy of models on AMZ Comp dataset.

## E  NUMERICAL RESULTS AND HYPERPARAMETERS

In order to facilitate future research to compare with our results, we share the accurate numeric results here in addition to the curves shown in the pictures of the Experimental section. We also share the experimental environment and the optimal hyperparameters we used to get the results in B.2.

### E.1  NUMERICAL RESULTS

#### E.1.1  NOISE RATE (NR)

Node Classification (NC)

Table 6: Cora - NR - NC

|        | 0.200     | 0.400     | 0.600     | 0.800     | 1.000     |
|--------|-----------|-----------|-----------|-----------|-----------|
| GCN    | 0.751     | 0.706     | 0.662     | 0.631     | 0.606     |
| SAGE   | 0.768     | 0.717     | 0.685     | 0.656     | 0.645     |
| GAT    | 0.713     | 0.668     | 0.626     | 0.605     | 0.603     |
| GIN    | 0.712     | 0.654     | 0.621     | 0.607     | 0.601     |
| GDC    | **0.814** | **0.806** | **0.799** | **0.784** | **0.783** |
| **BiGCN** | 0.802  | 0.785     | 0.770     | 0.745     | 0.734     |

Table 7: Citeseer - NR - NC

|        | 0.200     | 0.400     | 0.600     | 0.800     | 1.000     |
|--------|-----------|-----------|-----------|-----------|-----------|
| GCN    | 0.597     | 0.553     | 0.483     | 0.442     | 0.404     |
| SAGE   | 0.612     | 0.543     | 0.497     | 0.450     | 0.427     |
| GAT    | 0.564     | 0.457     | 0.405     | 0.371     | 0.346     |
| GIN    | 0.535     | 0.468     | 0.432     | 0.405     | 0.401     |
| GDC    | 0.617     | 0.575     | 0.548     | 0.520     | 0.511     |
| **BiGCN** | **0.626** | **0.580** | **0.561** | **0.531** | **0.516** |

Table 8: PubMed - NR - NC

|        | 0.200     | 0.400     | 0.600     | 0.800     | 1.000     |
|--------|-----------|-----------|-----------|-----------|-----------|
| GCN    | 0.550     | 0.466     | 0.434     | 0.422     | 0.385     |
| SAGE   | 0.579     | 0.489     | 0.439     | 0.438     | 0.420     |
| GAT    | 0.491     | 0.477     | 0.467     | 0.465     | 0.449     |
| GIN    | 0.568     | 0.505     | 0.482     | 0.490     | 0.478     |
| GDC    | 0.560     | 0.474     | 0.427     | 0.412     | 0.404     |
| **BiGCN** | **0.665** | **0.619** | **0.604** | **0.567** | **0.547** |

Table 9: Computers - NR - NC

|        | 0.200     | 0.400     | 0.600     | 0.800     | 1.000     |
|--------|-----------|-----------|-----------|-----------|-----------|
| GCN    | 0.837     | 0.832     | 0.810     | 0.839     | 0.830     |
| SAGE   | 0.846     | 0.831     | 0.836     | 0.840     | 0.838     |
| GAT    | 0.770     | 0.800     | 0.769     | 0.766     | 0.750     |
| GIN    | 0.420     | 0.402     | 0.402     | 0.394     | 0.406     |
| GDC    | 0.840     | 0.837     | 0.832     | 0.838     | 0.832     |
| **BiGCN** | **0.856** | **0.855** | **0.855** | **0.853** | **0.853** |

Table 10: Photos - NR - NC

|        | 0.200     | 0.400     | 0.600     | 0.800     | 1.000     |
|--------|-----------|-----------|-----------|-----------|-----------|
| GCN    | 0.913     | 0.908     | 0.907     | 0.905     | 0.894     |
| SAGE   | 0.910     | 0.903     | 0.900     | 0.901     | 0.904     |
| GAT    | 0.873     | 0.874     | 0.867     | 0.848     | 0.855     |
| GIN    | 0.342     | 0.315     | 0.333     | 0.304     | 0.306     |
| GDC    | 0.901     | 0.896     | 0.890     | 0.883     | 0.881     |
| **BiGCN** | **0.922** | **0.921** | **0.920** | **0.917** | **0.916** |

Link Prediction (LP)

Table 11: Cora - NR - LP

|        | 0.200     | 0.400     | 0.600     | 0.800     | 1.000     |
|--------|-----------|-----------|-----------|-----------|-----------|
| GCN    | 0.850     | 0.817     | 0.795     | 0.792     | 0.785     |
| SAGE   | 0.846     | 0.826     | 0.786     | 0.785     | 0.774     |
| GAT    | 0.848     | 0.817     | 0.781     | 0.785     | 0.767     |
| GIN    | 0.827     | 0.799     | 0.799     | 0.785     | 0.780     |
| GDC    | 0.872     | 0.860     | **0.853** | **0.847** | 0.840     |
| **BiGCN** | **0.887** | **0.875** | 0.851  | 0.845     | **0.843** |

Table 12: Citeseer - NR - LP

|        | 0.200     | 0.400     | 0.600     | 0.800     | 1.000     |
|--------|-----------|-----------|-----------|-----------|-----------|
| GCN    | 0.812     | 0.773     | 0.754     | 0.739     | 0.726     |
| SAGE   | 0.824     | 0.787     | 0.749     | 0.740     | 0.732     |
| GAT    | 0.807     | 0.765     | 0.747     | 0.738     | 0.741     |
| GIN    | 0.819     | 0.772     | 0.758     | 0.757     | 0.747     |
| DGC    | 0.808     | 0.779     | 0.758     | 0.764     | 0.756     |
| **BiGCN** | **0.867** | **0.836** | **0.812** | **0.800** | **0.804** |

Table 13: Pubmed - NR - LP

|        | 0.200     | 0.400     | 0.600     | 0.800     | 1.000     |
|--------|-----------|-----------|-----------|-----------|-----------|
| GCN    | 0.838     | 0.767     | 0.745     | 0.743     | 0.741     |
| SAGE   | 0.844     | 0.797     | 0.770     | 0.763     | 0.755     |
| GAT    | 0.840     | 0.789     | 0.775     | 0.777     | 0.778     |
| GIN    | 0.802     | 0.771     | 0.766     | 0.769     | 0.771     |
| GDC    | 0.839     | 0.801     | 0.780     | 0.769     | 0.760     |
| **BiGCN** | **0.875** | **0.846** | **0.825** | **0.811** | **0.803** |

Table 14: DBLP - NR - LP

|        | 0.200     | 0.400     | 0.600     | 0.800     | 1.000     |
|--------|-----------|-----------|-----------|-----------|-----------|
| GCN    | 0.901     | 0.879     | 0.868     | 0.860     | 0.854     |
| SAGE   | 0.899     | 0.879     | 0.868     | 0.857     | 0.856     |
| GAT    | 0.897     | 0.877     | 0.865     | 0.862     | 0.857     |
| GIN    | 0.890     | 0.879     | 0.875     | 0.872     | 0.872     |
| GDC    | 0.898     | 0.885     | 0.873     | 0.866     | 0.862     |
| **BiGCN** | **0.914** | **0.902** | **0.895** | **0.890** | **0.884** |

### E.1.2 NOISE LEVEL (NL)

Node Classification (NC)

Table 15: Cora - NL - NC

|        | 0.100 | 0.200 | 0.300 | 0.400 | 0.500 | 0.600 | 0.700 | 0.800 | 0.900 |
|--------|-------|-------|-------|-------|-------|-------|-------|-------|-------|
| GCN    | 0.792 | 0.749 | 0.704 | 0.643 | 0.606 | 0.585 | 0.572 | 0.558 | 0.530 |
| SAGE   | 0.791 | 0.758 | 0.720 | 0.677 | 0.646 | 0.628 | 0.616 | 0.603 | 0.599 |
| GAT    | 0.782 | 0.737 | 0.688 | 0.644 | 0.622 | 0.608 | 0.601 | 0.584 | 0.579 |
| GIN    | 0.734 | 0.686 | 0.659 | 0.617 | 0.610 | 0.580 | 0.595 | 0.581 | 0.577 |
| GDC    | **0.828** | **0.819** | **0.805** | **0.792** | **0.784** | **0.772** | **0.763** | **0.754** | **0.758** |
| **BiGCN** | 0.825 | 0.804 | 0.785 | 0.768 | 0.749 | 0.732 | 0.725 | 0.714 | 0.709 |

Table 16: Citeseer - NL - NC

|        | 0.100 | 0.200 | 0.300 | 0.400 | 0.500 | 0.600 | 0.700 | 0.800 | 0.900 |
|--------|-------|-------|-------|-------|-------|-------|-------|-------|-------|
| GCN    | 0.671 | 0.598 | 0.500 | 0.443 | 0.418 | 0.403 | 0.382 | 0.382 | 0.367 |
| SAGE   | 0.670 | 0.605 | 0.492 | 0.464 | 0.429 | 0.423 | 0.408 | 0.408 | 0.398 |
| GAT    | 0.620 | 0.534 | 0.476 | 0.436 | 0.405 | 0.374 | 0.333 | 0.309 | 0.293 |
| GIN    | 0.557 | 0.497 | 0.430 | 0.410 | 0.392 | 0.392 | 0.381 | 0.385 | 0.376 |
| GDC    | 0.656 | 0.613 | **0.555** | 0.523 | 0.515 | 0.506 | **0.497** | **0.489** | **0.486** |
| **BiGCN** | **0.677** | **0.619** | 0.552 | **0.524** | **0.521** | **0.514** | 0.482 | **0.489** | 0.476 |

Table 17: PubMed - NL - NC

|        | 0.100 | 0.200 | 0.300 | 0.400 | 0.500 | 0.600 | 0.700 | 0.800 | 0.900 |
|--------|-------|-------|-------|-------|-------|-------|-------|-------|-------|
| GCN    | 0.530 | 0.433 | 0.406 | 0.379 | 0.373 | 0.382 | 0.381 | 0.377 | 0.383 |
| SAGE   | 0.565 | 0.455 | 0.432 | 0.422 | 0.431 | 0.414 | 0.411 | 0.412 | 0.405 |
| GAT    | 0.537 | 0.445 | 0.434 | 0.430 | 0.435 | 0.436 | 0.426 | 0.434 | 0.433 |
| GIN    | 0.643 | 0.542 | 0.498 | 0.493 | 0.476 | 0.477 | 0.473 | 0.477 | 0.484 |
| GDC    | 0.550 | 0.440 | 0.415 | 0.409 | 0.388 | 0.389 | 0.400 | 0.393 | 0.389 |
| **BiGCN** | **0.688** | **0.593** | **0.554** | **0.530** | **0.526** | **0.512** | **0.517** | **0.505** | **0.506** |

Table 18: Computers - NL - NC

|        | 0.100 | 0.200 | 0.300 | 0.400 | 0.500 | 0.600 | 0.700 | 0.800 | 0.900 |
|--------|-------|-------|-------|-------|-------|-------|-------|-------|-------|
| GCN    | 0.844 | 0.846 | 0.845 | 0.846 | 0.825 | 0.805 | 0.836 | 0.831 | 0.829 |
| SAGE   | 0.838 | 0.845 | 0.837 | 0.837 | 0.827 | 0.826 | 0.836 | 0.825 | 0.825 |
| GAT    | 0.776 | 0.757 | 0.760 | 0.771 | 0.747 | 0.763 | 0.770 | 0.754 | 0.769 |
| GIN    | 0.409 | 0.388 | 0.411 | 0.399 | 0.400 | 0.404 | 0.413 | 0.424 | 0.410 |
| GDC    | 0.847 | 0.838 | 0.841 | 0.833 | 0.832 | 0.827 | 0.816 | 0.810 | 0.799 |
| **BiGCN** | **0.856** | **0.856** | **0.856** | **0.853** | **0.855** | **0.852** | **0.851** | **0.847** | **0.851** |

Table 19: Photos - NL - NC

|        | 0.100 | 0.200 | 0.300 | 0.400 | 0.500 | 0.600 | 0.700 | 0.800 | 0.900 |
|--------|-------|-------|-------|-------|-------|-------|-------|-------|-------|
| GCN    | 0.915 | 0.907 | 0.904 | 0.903 | 0.906 | 0.901 | 0.899 | 0.894 | 0.893 |
| SAGE   | 0.903 | 0.904 | 0.909 | 0.904 | 0.905 | 0.901 | 0.902 | 0.901 | 0.896 |
| GAT    | 0.879 | 0.881 | 0.871 | 0.868 | 0.849 | 0.845 | 0.840 | 0.770 | 0.740 |
| GIN    | 0.336 | 0.337 | 0.327 | 0.332 | 0.317 | 0.345 | 0.351 | 0.314 | 0.330 |
| GDC    | 0.901 | 0.896 | 0.896 | 0.887 | 0.885 | 0.877 | 0.876 | 0.863 | 0.859 |
| **BiGCN** | **0.925** | **0.923** | **0.920** | **0.919** | **0.919** | **0.916** | **0.908** | **0.906** | **0.902** |

Link Prediction (LP)

Table 20: Cora - NL - LP

|        | 0.100 | 0.200 | 0.300 | 0.400 | 0.500 | 0.600 | 0.700 | 0.800 | 0.900 |
|--------|-------|-------|-------|-------|-------|-------|-------|-------|-------|
| GCN    | 0.857 | 0.814 | 0.795 | 0.779 | 0.780 | 0.774 | 0.764 | 0.770 | 0.763 |
| SAGE   | 0.864 | 0.819 | 0.792 | 0.774 | 0.769 | 0.770 | 0.770 | 0.770 | 0.761 |
| GAT    | 0.850 | 0.814 | 0.790 | 0.785 | 0.775 | 0.763 | 0.768 | 0.765 | 0.767 |
| GIN    | 0.854 | 0.815 | 0.796 | 0.786 | 0.782 | 0.787 | 0.775 | 0.772 | 0.770 |
| GDC    | 0.877 | 0.862 | 0.845 | 0.844 | 0.833 | **0.839** | **0.839** | **0.832** | **0.833** |
| **BiGCN** | **0.892** | **0.865** | **0.860** | **0.844** | **0.836** | 0.837 | 0.830 | 0.827 | **0.833** |

Table 21: Citeseer - NL - LP

|        | 0.100 | 0.200 | 0.300 | 0.400 | 0.500 | 0.600 | 0.700 | 0.800 | 0.900 |
|--------|-------|-------|-------|-------|-------|-------|-------|-------|-------|
| GCN    | 0.807 | 0.760 | 0.743 | 0.736 | 0.725 | 0.726 | 0.713 | 0.714 | 0.713 |
| SAGE   | 0.815 | 0.762 | 0.730 | 0.737 | 0.734 | 0.723 | 0.715 | 0.719 | 0.718 |
| GAT    | 0.799 | 0.760 | 0.746 | 0.736 | 0.726 | 0.740 | 0.725 | 0.724 | 0.722 |
| GIN    | 0.838 | 0.780 | 0.748 | 0.744 | 0.744 | 0.731 | 0.730 | 0.734 | 0.736 |
| GDC    | 0.820 | 0.777 | 0.760 | 0.743 | 0.747 | 0.744 | 0.745 | 0.738 | 0.743 |
| **BiGCN** | **0.874** | **0.836** | **0.811** | **0.799** | **0.797** | **0.795** | **0.792** | **0.791** | **0.780** |

Table 22: Pubmed - NL - LP

|        | 0.100 | 0.200 | 0.300 | 0.400 | 0.500 | 0.600 | 0.700 | 0.800 | 0.900 |
|--------|-------|-------|-------|-------|-------|-------|-------|-------|-------|
| GCN    | 0.821 | 0.774 | 0.753 | 0.749 | 0.741 | 0.739 | 0.735 | 0.738 | 0.738 |
| SAGE   | 0.813 | 0.774 | 0.761 | 0.753 | 0.751 | 0.747 | 0.749 | 0.750 | 0.752 |
| GAT    | 0.820 | 0.779 | 0.773 | 0.770 | 0.770 | 0.770 | 0.770 | 0.770 | 0.772 |
| GIN    | 0.806 | 0.782 | 0.772 | 0.768 | 0.769 | 0.770 | 0.773 | 0.775 | 0.771 |
| GDC    | 0.828 | 0.785 | 0.758 | 0.755 | 0.753 | 0.750 | 0.745 | 0.745 | 0.748 |
| **BiGCN** | **0.841** | **0.814** | **0.806** | **0.804** | **0.801** | **0.799** | **0.800** | **0.800** | **0.796** |

Table 23: DBLP - NL - LP

|        | 0.100 | 0.200 | 0.300 | 0.400 | 0.500 | 0.600 | 0.700 | 0.800 | 0.900 |
|--------|-------|-------|-------|-------|-------|-------|-------|-------|-------|
| GCN    | 0.902 | 0.881 | 0.866 | 0.857 | 0.851 | 0.847 | 0.843 | 0.842 | 0.841 |
| SAGE   | 0.900 | 0.876 | 0.863 | 0.859 | 0.858 | 0.856 | 0.851 | 0.850 | 0.849 |
| GAT    | 0.899 | 0.880 | 0.868 | 0.863 | 0.860 | 0.856 | 0.854 | 0.853 | 0.850 |
| GIN    | 0.896 | 0.885 | 0.878 | 0.874 | 0.872 | 0.869 | 0.865 | 0.867 | 0.865 |
| GDC    | 0.904 | 0.883 | 0.871 | 0.865 | 0.859 | 0.854 | 0.853 | 0.850 | 0.847 |
| **BiGCN** | **0.916** | **0.900** | **0.894** | **0.890** | **0.884** | **0.883** | **0.883** | **0.880** | **0.881** |

### E.1.3 STRUCTURE MISTAKES (SM)

Node Classification (NC)

Table 24: Cora - SM - NC

|  | 0.001 | 0.003 | 0.005 | 0.007 | 0.009 | 0.011 | 0.013 | 0.015 |
|---|---|---|---|---|---|---|---|---|
| GCN | 0.693 | 0.611 | 0.564 | 0.527 | 0.507 | 0.486 | 0.468 | 0.468 |
| SAGE | 0.697 | 0.606 | 0.558 | 0.519 | 0.515 | 0.500 | 0.483 | 0.473 |
| GAT | 0.678 | 0.565 | 0.509 | 0.458 | 0.435 | 0.400 | 0.356 | 0.342 |
| GIN | 0.529 | 0.302 | 0.261 | 0.247 | 0.242 | 0.244 | 0.275 | 0.261 |
| GDC | **0.723** | 0.659 | 0.634 | 0.619 | 0.611 | 0.604 | 0.598 | 0.599 |
| **BiGCN** | 0.720 | **0.682** | **0.659** | **0.645** | **0.643** | **0.637** | **0.631** | **0.635** |

Table 25: Citeseer - - SM - NC

|  | 0.001 | 0.003 | 0.005 | 0.007 | 0.009 | 0.011 | 0.013 | 0.015 |
|---|---|---|---|---|---|---|---|---|
| GCN | 0.521 | 0.446 | 0.409 | 0.378 | 0.362 | 0.351 | 0.347 | 0.329 |
| SAGE | 0.520 | 0.446 | 0.411 | 0.375 | 0.369 | 0.363 | 0.336 | 0.337 |
| GAT | 0.498 | 0.397 | 0.344 | 0.299 | 0.282 | 0.272 | 0.254 | 0.231 |
| GIN | 0.397 | 0.221 | 0.211 | 0.210 | 0.201 | 0.204 | 0.206 | 0.205 |
| GDC | 0.529 | 0.515 | 0.515 | 0.511 | 0.516 | 0.514 | 0.518 | 0.526 |
| **BiGCN** | **0.601** | **0.577** | **0.565** | **0.567** | **0.565** | **0.565** | **0.562** | **0.567** |

Table 26: PubMed - SM - NC

|  | 1e-5 | 2.5e-5 | 5e-5 | 7.5e-5 | 1e-4 | 2.5e-4 | 5e-4 | 7.5e-4 | 1e-3 |
|---|---|---|---|---|---|---|---|---|---|
| GCN | 0.770 | 0.760 | 0.745 | 0.730 | 0.719 | 0.683 | 0.659 | 0.647 | 0.640 |
| SAGE | 0.770 | 0.760 | 0.739 | 0.731 | 0.721 | 0.679 | 0.659 | 0.645 | 0.638 |
| GAT | 0.772 | 0.755 | 0.734 | 0.721 | 0.706 | 0.665 | 0.641 | 0.633 | 0.610 |
| GIN | 0.774 | 0.768 | 0.758 | 0.754 | 0.742 | 0.700 | 0.577 | 0.497 | 0.442 |
| GDC | 0.766 | 0.755 | 0.742 | 0.730 | 0.723 | 0.690 | 0.692 | 0.707 | 0.713 |
| **BiGCN** | **0.778** | **0.770** | **0.761** | **0.751** | **0.746** | **0.728** | **0.721** | **0.724** | **0.725** |

Table 27: Computers - SM - NC

|  | 0.001 | 0.003 | 0.005 | 0.007 | 0.009 | 0.011 | 0.013 | 0.015 |
|---|---|---|---|---|---|---|---|---|
| GCN | 0.782 | 0.723 | 0.632 | 0.548 | 0.519 | 0.472 | 0.471 | 0.457 |
| SAGE | 0.779 | 0.708 | 0.660 | 0.562 | 0.531 | 0.496 | 0.471 | 0.456 |
| GAT | 0.663 | 0.597 | 0.524 | 0.530 | 0.499 | 0.432 | 0.443 | 0.429 |
| GIN | 0.417 | 0.395 | 0.385 | 0.389 | 0.386 | 0.385 | 0.385 | 0.385 |
| GDC | 0.776 | 0.744 | 0.711 | 0.690 | 0.678 | 0.684 | 0.677 | 0.682 |
| **BiGCN** | **0.822** | **0.795** | **0.775** | **0.764** | **0.756** | **0.756** | **0.749** | **0.749** |

Table 28: Photos - SM - NC

|  | 0.001 | 0.003 | 0.005 | 0.007 | 0.009 | 0.011 | 0.013 | 0.015 |
|---|---|---|---|---|---|---|---|---|
| GCN | 0.838 | 0.667 | 0.668 | 0.554 | 0.561 | 0.525 | 0.520 | 0.449 |
| SAGE | 0.855 | 0.815 | 0.762 | 0.754 | 0.710 | 0.638 | 0.592 | 0.478 |
| GAT | 0.820 | 0.681 | 0.512 | 0.432 | 0.376 | 0.372 | 0.356 | 0.319 |
| GIN | 0.266 | 0.283 | 0.261 | 0.262 | 0.261 | 0.260 | 0.267 | 0.255 |
| GDC | 0.871 | 0.843 | 0.818 | 0.787 | 0.764 | 0.748 | 0.748 | 0.734 |
| **BiGCN** | **0.902** | **0.880** | **0.867** | **0.856** | **0.851** | **0.844** | **0.838** | **0.833** |

## E.2 ADDITIONAL IMPLEMENTATION DETAILS AND HYPER-PARAMETER SETTING

All implementations for both node classification and link prediction are based on PyTorch 1.2.0 and Pytorch Geometric[1]. All experiments based on PyTorch are running on one NVIDIA GeForce RTX 2080 Ti GPU using CUDA. The experimental datasets are taken from the PyTorch Geometric platform. We tune our hyperparameters for each model using validation data and listed the final optimal setting in the following tables. To accelerate the tedious process of hyper-parameters tuning, we set $\frac{2\lambda_1}{1+p} = \frac{2\lambda_2}{1+p} = \lambda$ and choose different hyper-parameter $p$ for different datasets.

### E.2.1 NODE CLASSIFICATION

Table 29: Hyper-parameters of BiGCN in Node Classification

| Cases | Dataset | $p$ | $\lambda$ | $k$ | Hidden dimension | Layer | Dropout | lr |
|---|---|---|---|---|---|---|---|---|
| Noise rate | Cora | 3 | 1.8 | | | | | |
| | Citeseer | 3 | 1.8 | | | | | |
| | PubMed | 3 | 1.8 | 2 | 16 | 2 | 0.5 | 0.01 |
| | Comp | 2.5 | 1.0 | | | | | |
| | Photos | 1.5 | 0.8 | | | | | |
| Noise level | Cora | 3 | 1.8 | | | | | |
| | Citeseer | 3 | 1.8 | | | | | |
| | PubMed | 3 | 1.8 | 2 | 16 | 2 | 0.5 | 0.01 |
| | Comp | 2.5 | 1.0 | | | | | |
| | Photos | 1.5 | 0.8 | | | | | |
| Structure mistakes | Cora | 0.1 | 0.8 | | | | | |
| | Citeseer | 0.05 | 0.8 | | | | | |
| | PubMed | 0.1 | 0.8 | 2 | 16 | 2 | 0.5 | 0.01 |
| | Comp | 0.1 | 1.0 | | | | | |
| | Photos | 0.1 | 1.0 | | | | | |

---

[1]https://github.com/rusty1s/pytorch_geometric

### E.2.2 LINK PREDICTION

Table 30: Hyper-parameters of BiGCN in Link Prediction

| Cases | Dataset | $p$ | $\lambda$ | $k$ | Hidden dimension | Layer | Dropout | lr |
|---|---|---|---|---|---|---|---|---|
| Noise rate | Cora | 8.5 | 1.2 | | | | | |
| | Citeseer | 8.5 | 1.2 | | | | | |
| | PubMed | 8.5 | 1.2 | 2 | 32 | 2 | 0.5 | 0.01 |
| | DBLP | 8.5 | 1.2 | | | | | |
| Noise level | Cora | 8.5 | 1.2 | | | | | |
| | Citeseer | 8.5 | 1.2 | | | | | |
| | PubMed | 8.5 | 1.2 | 2 | 32 | 2 | 0.5 | 0.01 |
| | DBLP | 8.5 | 1.2 | | | | | |

