# OpenReview forum: "BiGCN: A Bi-directional Low-Pass Filtering Graph Neural Network"
_ICLR.cc/2021/Conference — Reject_

### Official Review · AnonReviewer1 · 2020-10-19
**The experiment seems to show some evidence for the proposed idea but the methodology itself is doubtful**

**Rating:** 4
**Confidence:** 5

**Review:**

Summary:

The authors proposed a to apply “low pass filtering” on both node and feature domain.

Pros:
1.	Interesting idea on trying to define and apply low pass filter on feature domain
2.	The experiment results sort of validates the proposed idea.

Cons:
1.	The definition of “feature graph” seems questionable. Moreover, the authors propose to learn the feature graph $L_2$ in the actual implementation. Compare to node domain where the graph topology is given by data, this asymmetric is very weird to me.
2.	The motivation of applying additional filtering on feature domain seems redundant. Doesn’t the weight matrix $W^{(l)}$ in each layer work as the feature transformation already?

Detailed comments:

The main weakness of the paper would be its motivation. Indeed, as the author claim, there might be some correlations among features. However, the weight matrices $W^{(l)}$ already serve as feature transformation for the $l^{\text{th}}$ GCN layer. Why do we need additional “low pass filtering” in feature domain? If the methodology of this paper is correct, then why we still need the additional weighted matrix $W^{(l)}$ as in equation (11)? Also, even if we are given with some feature graph, why the low pass filtering defined with respect to this feature graph is reasonable? Note that the low pass filtering on graph is strongly related to the “Homophily principle”[1] which has been verified through extensive studies in network science community. However, it is not clear that the same conclusion will hold for the newly defined feature graph. Lastly, even in node domain the homophily principle is known to be invalid for some real graphs [2]. I think the authors should explain and discuss in more detail on this.

Consider the case were we only have $1$ node with $2$ features $Y = [y_1,y_2]$. Furthermore, consider the node label function $f(y_1,y_2) = sign(y_2-y_1-1)$. Also, let $y_2 = 11y_1$ and $y_1$ is generated uniformly in range $[0,1]$. In this case, we can see that graph topology in node domain is irrelevant and we can focus on the feature domain. Also, the label function is positive with high probability (0.9) and the correlation coefficient between $y_1,y_2$ is $1$. According to the definition of feature graph (Section 4.2, page 4), the node $y_1$ and $y_2$ are linked and the corresponding low pass filter is just averaging $\frac{y_1+y_2}{2}$ (up to a constant factor). Hence, the output feature matrix becomes closer to $[\frac{y_1+y_2}{2},\frac{y_1+y_2}{2}]$ as $\lambda_2$ larger (equation (4)). Note that even if we know the optimal label function $f$, now we are more likely to predict negative label (with probability 1 when $\lambda_2\rightarrow\infty$). In this example, it seems like applying low pass filtering in feature domain actually hurt the label prediction performance. What I try to convey here is that merely considering the correlation in node feature is insufficient. We have to also take the label correlation into account. Maybe I misunderstand something here but I hope the authors can elaborate on this.

Even if we accept the idea that we really need some low pass filtering in feature domain, the definition of “low pass filtering” given by authors seems problematic to me. In node domain, since we are given the graph topology directly from data, we are able to define “frequency” based on the spectrum of graph Laplacian. In contrast, we are not given “feature graph” that characterizes the correlation between features. The author proposed to learn the graph Laplacian of the underlying feature graph $L_2$ but I doubt the correct $L_2$ can be learnt. Furthermore, the authors use different $L_2$ for each layer which is a bit counterintuitive to me. It is weird that the underlying “feature graph” topology will change across layers. Note that the graph topology remains the same for all layers in GNNs. Finally, instead of using this convoluted design, why not just apply multilayer perceptron (MLP) to do the job? It is known that MLP can approximate arbitrary function and thus should be able to approximate the underlying “low pass filter” for feature graph. I think the authors need to explain better on why the proposed design is necessary under their motivation.

As a final comment, I think the experimental results seems interesting. Indeed, the noise is injected artificially but it kind of verifies the idea of low pass filtering in feature domain for Gaussian noise case. The paper would be greatly improved if the concerns above can be addressed.

Reference:

[1] “Birds of a feather: Homophily in social networks.,” McPherson et al., Annual review of sociology, 2001.

[2] “Geom-GCN: Geometric Graph Convolutional Networks,” Pei et al., ICLR 2020.

---

> ### Author Response · Authors · 2020-11-25
> **Response to your comments. Thanks for your feedback.**
>
> We thank the reviewer for the helpful and constructive comments. We respond to each question as follows:
>
> **Q1.The weight matrices W(l)  already serve as feature transformations for the lth GCN layer. Why do we need additional “low pass filtering” in the feature domain?**
>
> ANS: Feature transformation is obviously different from filtering. Considering that if there is noise, the representations after transformation will still include the noise, while the filtering can remove the noise. In addition, by learning a latent feature interaction graph, we also added more information than using a weight matrix.
> To further illustrate this point, let us take the well-known paper LDS [1] as an example. When we do not have graphs but just the node features, we can use an MLP to predict the node labels; but if we learn a latent graph and then use a GNN, the performance will be much better. Note that MLP is a transformation, while learning the latent graph and then using a GNN is low-pass filtering.
>
> **Q2.MLP can approximate arbitrary functions and thus should be able to approximate the underlying “low pass filter” for the feature graph.**
>
> ANS: “MLP can approximate arbitrary functions”  is just theoretically true, but it is well known that in practice MLP is limited to the model capacity and generalizability, so actually it cannot always achieve the effect that one expects. The LDS paper[1] that we mentioned is a very good illustration that MLP is not enough.
>
> **Q3.Homophily principle in feature graph.**
>
> ANS: Homophily principle holds for social networks but not for all graphs (for example, a protein-protein-interaction graph obviously does not follow this principle). In our model, the latent feature graph is actually learned from the data, so we believe it should fit the data well.
>
> **Q4.Consider the case were we only have node with features Y=[y1,y2]**
>
> ANS: The definition of the feature adjacency matrix  in section 4.2 is just an example to show our motivation. As we explained in the paper, in our model the $L_2$ is actually not heuristically defined but learned from the data. Your example is a very interesting case, but please note that the two features do not necessarily have a connection. It is quite possible that our model learns an $L_2$ with very low weights for the links between $y_1$ and $y_2$ or even no links. Moreover, we do not only have the filtering phase to solve equation(9), but also have another learnable parameter W^l in equation (11). As we said in Appendix C, the $L_2$ can even be an identity matrix and make the model degrade to a GCN-like model without any feature correlations. So the given example cannot defeat our model. To make the motivation and our model more clear, we change the example in section 4.2 in the revised paper.
>
>
> **Q5.The author proposed to learn the graph Laplacian of the underlying feature graph L2  but I doubt the correct L2 can be learned.**
>
> ANS: There is no ground-truth for $L_2$, i.e. no “correct” $L_2$. If it works in experiments, it can be correct. It can be just regarded as a parameter in our model. The learned $L_2$ is the best parameter to fit our model and the data. And our experiments show that it works.  Let us take LDS [1] again as an example, they do not learn a “correct” graph, but the learned latent graph helps with the performance and make the model better. LDS is not the only pape to demonstrate that, learning a latent graph is proved useful in many previous works (e.g. [2][3]).
>
> **Q6.Using different L2 for each layer.**
>
> ANS: Using different $L_2$ for each layer is more flexible, more reasonable and more practical. The features are transformed in each layer such that the semantic information of feature dimensions may be quite different, therefore the latent correlation relationship will change substantially as well. In addition, if the hidden units are usually different from layer to layer, i.e. the feature dimension is changed, it implies that the size of $L_2$ must be changed accordingly. In this case, $L_2$ has no way to be the same for each layer.
>
>
>
> [1] Learning Discrete Structures for Graph Neural Networks. Franceschi et al. 2019.
> [2] Neural Relational Inference for Interacting Systems. Kipf et al. 2018.
> [3] Iterative Deep Graph Learning for Graph Neural Networks. Chen et al. 2020.

---

### Official Review · AnonReviewer2 · 2020-10-27
**A good paper that introduces a GCN that is more robust to noisy data than some models in the literature**

**Rating:** 6
**Confidence:** 3

**Review:**

The manuscript introduces a graph convolutional layer based on the optimal solution of the minimization problem of recovering the true graph signal given a noisy observation. Moreover, the authors propose to consider the original graph structure information together from the latent correlation between nodes features. The proposed solution is interesting and the paper is well written. Only section 4.2 turns out to be a bit difficult to read. My suggestion is to explain a bit more deeply the concept of latent features connection graph already in the introduction section.  Another point that it is not clear to me, is related to the fact that the authors highlight several times the connection between the proposed convolution and the low pass filter concept. In (NT & Maehara, 2019) the authors state that many papers on graph neural networks iteratively multiply the adjacency matrix and this operation corresponds to a low-pass filter. Therefore it is not clear to me the reason why the proposed convolution differs from the convolutions in the literature on this aspect.

For what concerns the experimental results it is interesting to notice that the BiGCN is significantly more robust than the models considered in the comparison it is not clear how the proposed model performs on clean data. The results reported in appendix D.2 show comparable performance with the other considered models. The problem is that methodology used in the comparison in my opinion is not completely fair. Indeed the authors in Appendix D state that they used the same hyperparameters (lr,  weight decay, and dropout) for all benchmarks and baselines, while the epoch was chosen based on the validation set. Honestly, I think that in this way the authors introduce a bias on the comparison that could affect the results. In this regards in appendix E2 the authors state “We tune our hyperparameters for each model using validation data” (to me it is not clear what the authors meaning with “our hyperparameters”) but the information about how the validation process is performed is missing (grid-search/random search?). It is also not clear if this procedure is used also for the baseline models (for instance, how the hidden dimension had been chosen?).

In the last part of section 5, the authors discuss the “Structure mistake case”. Since the authors consider also the semi-supervised node classification task, where dataset like Cora, Pubmed, and Citeser basically use a very small sub-graph as a training set, it is interesting to know how much the introduced incorrect interaction relationships among the nodes, interest the training/test/validation sets. In the paragraph about the “Noise rate case” a more deep discussion about the results in the Cora dataset should be inserted. In particular to me, it is not clear why on Cora the GCN outperforms the BiGCN (in Citeseer the performance is very similar), while in Pubmed GCN archives one of the lower accuracy. Is there any difference in the data that justify this behavior?

In my opinion, in the set of models considered in the comparison, the authors should also consider ARMA, which has the advantage to be more robust than the other models in the literature (as reported in section 2.2). I suggest also considering in the comparison some more novel models proposed in the literature e.g. the ones defined in “Break the Ceiling: Stronger Multi-scale Deep Graph Convolutional Networks” by Luan et al. (2019),  that obtained very interesting results in the considered datasets on semi-supervised nodes classification.

Minor comments:
-In section 3 the symbol \lambda_i is used without been introduced it in advance;
-in sections 3/4 the authors use the same symbol (L) for the  Laplacian Matrix and for the normalized Laplacian matrix.
-please insert the complete appendix name when it is referred. For instance, in the last part of section 4, the authors state “ More technical details are added in Appendix”, without specifying which is the appendix referred to in this context.

---

> ### Author Response · Authors · 2020-11-25
> **Response to your comments. Thanks for your feedback and support.**
>
> **1.In (NT & Maehara, 2019) the authors state that many papers on graph neural networks iteratively multiply the adjacency matrix and this operation corresponds to a low-pass filter. Therefore it is not clear to me the reason why the proposed convolution differs from the convolutions in the literature on this aspect.**
>
> ANS: Multiplying the adjacency matrix is not the only way to design the low-pass graph fitler. The laplacian smoothing term used in our paper is another valid way and also another aspect to understand GNN. The reason why we use it is that  it can provide an easy way to add the latent feature interactions to make feature-wise filtering, as shown in the paper.
>
> **2.The methodology used in the comparison in my opinion is not completely fair.**
>
> ANS: For all the baselines and our models, the chosen of the epoch is consistent: using the  validation set. We tuned our hyperparameters listed in Appendix E.2 using a random search scheme. For baselines, we used the hyperparameters reported in the original papers.
>
> **3.In the last part of section 5, the authors discuss the “Structure mistake case”. It is interesting to know how much the introduced incorrect interaction relationships among the nodes, interest the training/test/validation sets. For the noise rate case, it is not clear why on Cora the GCN outperforms the BiGCN (in Citeseer the performance is very similar), while in Pubmed GCN archives one of the lower accuracy. Is there any difference in the data that justify this behavior?**
>
> ANS: For each run, the incorrect interaction relationships are randomly introduced to the whole graph. Therefore, it is hard to determine how much they are among the training / test / validation sets. For the noise rate case, it is GDC ( not GCN) that outperforms our model on Cora and archives almost the lowest accuracy on Pubmed. To find the difference between this two dataset, we apply Label Propagation algorithm and GCN with no features (i.e. take the identity matrix as input) for node classification:
>
> $\qquad$$\qquad$ GCN$\qquad$LP	$\quad$ GCN (no features)
>
> Cora	$\qquad$ 0.818	$\quad$  0.68 $\qquad$ 	0.603
>
> Pubmed$\quad$0.789 $\quad$   0.63	$\qquad$  0.480
>
> The experimental results show that, compared with that of Pubmed, the graph structure of Cora provides more valid information for classification. In other words, features of Pubmed have more impacts on performance and BiGCN may learn more latent information for classification.
>
> **4.More baselines.**
>
> ANS: Despite ARMA has a more advanced spectral filter, it still deals with the graph filtering in the original graph while ignoring the feature-wise filtering. In our experiments, we believe that we already have enough baselines, many of which can be regarded as advanced spectral filtering methods (e.g. GDC). Considering the amount of experiments that we have already done, we do not think adding another baseline is helpful.

---

### Official Review · AnonReviewer4 · 2020-10-28
**Official Blind Review #4**

**Rating:** 5
**Confidence:** 3

**Review:**

This paper proposed a new graph convolutional network. It considers not only the original graph structure information but also the latent correlations between features, resulting in a graph neural network as a bi-directional low-pass filter. The new filter is derived using the alternating direction method of multipliers (ADMM) algorithm. Experiments show the new model's denoising performance is better than previous models.

Pros:

- The idea of using feature correlations in graph neural networks is interesting. This technique seems to improve the model’s denoising performance.

- The new model outperforms previous graph neural networks on most of the benchmark datasets in different noisy settings.

Cons:

- The ADMM algorithm is applied to solve the optimization problem (4). However, the convergence of this algorithm with Taylor approximations is not provided in this paper. In fact, there are simpler algorithms with convergence guarantees that can be applied to solve the Sylvester equation (5), e.g., the gradient based iterative algorithm in "Gradient based iterative algorithms for solving a class of matrix equations" by Feng Ding and Tongwen Chen.

- All graph neural networks are compared when they have two-layers. However, in the new model, $L_2$ is a learnable symmetric matrix, which makes the new model more complex compared with other models. This comparison might be unfair to other models.

Because of the above reasons, I am leaning towards rejection. Below are some additional comments.

1. On page 3, "the smoothness of a graph signal $x$ can be measure through..." should be "... measured ...".

2. The notations in Figure 1 are not consistent with the notations in the paragraphs. It is better to make them consistent to avoid confusion.

3. On page 4, "$L$ is the (normalized) Laplacian matrix." It is better to give the definition of the normalized Laplacian matrix for completeness.

4. On page 4, section 4.2, $L_2$ and $L'$ are used interchangeably. It is better to make them consistent.

5. On page 5, "all of them require Schur decomposition which including Householder transforms..." should be "... includes..."

6. On page 5, equation (6), why is the L2 term evenly split into $f(Y_1)$ and $g(Y_2)$? What will happen if we set $f(Y_1) = ||Y_1-F||_F^2 + \lambda_1 trace(Y_1^TL_1Y_1)$ and $g(Y_2) = \lambda_2 trace(Y_2L_2Y_2^T)$?

7. On page 5, "... by choosing appropriate hyper-parameters $p$..." The matrix $L_2$ is learnable in the model, which makes it difficult to guarantee that the eigenvalues of $2\lambda_2L_2/(1+p)$ all fall into $[-1,1]$.

8. Please include the initializations of the ADMM algorithm.

9. On page 6, "we update $Y_1$ by appling the... then update $Y_2$ by appling the..." should be "... applying... applying..."

10. On page 6, "It also explains that BiGCN is more expressive that single-direction los-pass filtering GCNs" should be "... than..."

---

> ### Author Response · Authors · 2020-11-25
> **Response to your comments. Thanks for your feedback.**
>
> We thank the reviewer for the helpful and constructive comments. We respond to each question as follows:
>
> **Q1.The convergence of this algorithm with Taylor approximations is not provided in this paper.**
>
> ANS: In this paper, using ADMM to solve the convex optimization problem (4) is just a motivation to design the proposed neural network, BIGCN. So it does not impact a lot if we use some approximations in the model design (even if the approximation does not converge); and the experimental results also demonstrate that the approximation works. Actually, using approximations in the model design of GNNs have been quite common, e.g. APPNP, GCN.
>
> **Q2.$L_2$ is a learnable symmetric matrix, which makes the new model more complex compared with other models. This comparison might be unfair to other models.**
>
> ANS: We respectfully disagree with this comment. First, adding the model capacity is a fair way to design a new model. For example, deeper GNN models (such as JK-Net) obviously make the model more complex, no one says it is unfair; using multi-head attentions in GAT also increases the complexity, no one says it is unfair. We use the same input as other methods, and the $L_2$ matrix is also learned from the given data. It is indeed a fair comparison.
> For the number of layers of baselines, using two-layer networks is a well-known default setting for these models since it can already achieve a good performance (as indicated in GCN).
>
> **Q3.Please include the initializations of the ADMM algorithm.**
>
> ANS: The initializations of $Y_1$ and $Y_2$ are the feature matrix and $Z$ is a zero matrix.
>
> **Q4.On page 5, equation (6), why is the $L_2$ term evenly splitted into $f(Y_1)$ and $g(Y_2)$? What will happen if we set $f(Y_1) = ||Y_1-F||_F^2 + \lambda_1 trace(Y_1^TL_1Y_1)$ and $g(Y_2) = \lambda_2 trace(Y_2^TL_2Y_2)$?**
>
> ANS: That is a good question. To solve one optimization problem, there can be different splitting methods. Your mentioned way is also valid, but note that $Y_1$ and $Y_2$ should be the same when converged, so we think splitting in a symmetric way may make the convergence faster.
>
> **Q5.On page 5, "... by choosing appropriate hyper-parameters p..." The matrix $L_2$ is learnable in the model, which makes it difficult to guarantee that the eigenvalues of $2\lambda_2/(1+p)L_2$ all fall into $[-1,1]$.**
>
> ANS: We do not directly learn $L_2$ but a parameter for $L_2$. The $L_2$ is defined always as a valid Lapalician matrix from this parameter as we showed in section 5.2. So the eigenvalues of $2\lambda_2/(1+p)L_2$ will always fall into $[-1,1]$.

---

### Official Review · AnonReviewer3 · 2020-10-29
**Regularizing GNN on feature graph is interesting, but more careful evaluation is needed.**

**Rating:** 5
**Confidence:** 3

**Review:**

Pros:
- In addition to encouraging the node embeddings to be smooth over the graph space, the paper further regularize the embeddings to be smooth among different features.
- The method outperforms comparison methods when data is polluted with three types of random noise.
- The paper is well-organized and clearly written. To my best knowledge, the method is technically sound.

Cons:
- Compared with previous methods, the proposed method only achieves comparable accuracy performance on the real-world datasets.
- The Gaussian noise is too weak to evaluate the robustness of the method.
- It is better to give an analysis of the computational cost of Equ.(11) and provide an empirical speed comparison.

---

> ### Author Response · Authors · 2020-11-25
> **Response to your comments. Thanks for your feedback.**
>
> We thank the reviewer for the helpful and constructive comments. We respond to each question as follows:
>
> **1.Compared with previous methods, the proposed method only achieves comparable accuracy performance on real-world datasets.**
>
> As we showed in Appendix D.2 table4 and table5, our model outperformed other baselines on most of the benchmarks although the improvement is small for clean data. Actually, it is understandable: the benchmarks are quite clean and their features are redundant and some of them are alternatives which provide sufficient feature information. However, for noisy data, when feature information or structure information is unreliable, the extra correlation information is particularly important as shown in experimental results on artificial noisy data.  A mass of experiential results showed that the previous methods almost collapsed when introducing noise to the data. The absolute superiority of our model strongly and credibly demonstrated our impressive denoising capacity and robustness.
>
> **2.The Gaussian noise is too weak to evaluate the robustness of the method.**
>
> Considering that our motivation is to filter the noise in graphs and features, Gaussian noise is the most practical case in the real world, and it has been used in previous papers such as BayesianGCN. We want to emphasize that our model is not a defense method against adversarial attacks, and robustness does not only mean that.
>
> **3.It is better to give an analysis of the computational cost of Equ. (11) and provide an empirical speed comparison.**
>
> Here we provide the average running time (in seconds) of run of baseline and our model on the node classification task:
>
> $\qquad\\qquad\$ GCN 	|SAGE |$\quad\$GAT	|GIN	|$\quad\$GCN_PPR|BIGCN|
>
> Cora 	 $\qquad\$        15.3	$\quad\$  8.4 $\quad\$	50.1	 $\quad\$ 3.4 $\qquad\$ 18.6$\qquad\$44.9
>
> Citeseer    $\quad\$10.1$\quad\$  19.6  $\quad\$  34.5   $\quad\$3.3       $\qquad\$    16.5        $\quad\$     134.4
>
> PubMed      $\quad\$8.5   $\quad\$      21.6   $\quad\$ 38.1  $\quad\$   9.4    $\qquad\$      279.3$\quad\$410.8
>
> Comp$\qquad\$ 12.0$\quad\$70.5  $\quad\$ 90.1  $\quad\$292.9 $\quad\$ 104.8$ \quad\$199.2
>
> Photos   $\quad\$      13.2     $\quad\$    45.1   $\quad\$ 63.0  $\quad\$    4.9    $\qquad\$       47.8     $\quad\$         84.6
>
> Note that the contribution of our method is not for accelerating the graph neural networks, the comparison of running time is never required for such papers. Although BIGCN is slower than GCN, its running time is still acceptable. It does not impact the merit of BIGCN considering the noise filtering benefit, especially when the noise rate is high.

---

### Decision · Program_Chairs · 2021-01-07
**Final Decision**

**Decision:**

Reject

**Comment:**

From the positive side the problem addressed by the paper could be of potential interest in the case there is noise in the features associated to each node of the graph. The paper is mostly well written and clear. The proposed approach is based on solid mathematical grounds.

On the other hand there are concerns about:

i) motivation: it is not clear how significant the proposed approach is since the authors were not able to clearly highlight the advantages with respect to the standard approach where already the weight matrix (via learning) can play the role of a low-pass filter for node’s features. Maybe the main advantage is given by the fact that the network does not have to learn a low-pass filter, however this needs a better clarification;

ii) suggested approach: the authors are using an approach that seems to be more complex with respect to simpler ones already proposed in literature and not mentioned in the paper. In addition to that, the simpler approaches have convergence guarantees that have not been proved for the proposed approach;

iii) significance of the experimental results: the obtained experimental results are obtained by using a model with more parameters with respect to the baselines. Comparisons versus baselines with a similar number of parameters are necessary to have a fair assessment of the merits of the proposed approach.